# Meso-Scale Failure of Freezing–Thawing Damage of Concrete under Uniaxial Compression

**Zheng Si \*, Xiaoqi Du, Lingzhi Huang and Yanlong Li**

State Key Laboratory Base of Eco-Hydraulic Engineering in Arid Area (Xi'an University of Technology), Xi'an University of Technology, Xi'an 710048, China; 2170420092@stu.xaut.edu.cn (X.D.); hlzsz@126.com (L.H.); liyanlong@xaut.edu.cn (Y.L.)

\* Correspondence: sizheng@xaut.edu.cn



**Featured Application: Model established by discrete element method can predict damage degree of freeze-thaw concrete.**

**Abstract:** In this study, discrete element PFC3D software was used to simulate the uniaxial compression of C30 concrete specimen models after freezing–thawing damage. Based on the thickness of the freezing–thawing damage layer of the concrete and the uniaxial compression test, the meso-parameters of cement mortar in concrete and the meso-parameters of weak interface between the cement mortar and coarse aggregate under different freezing–thawing cycles were calibrated. The axial compression deformation produced in the experiment was compared with the axial displacement of particles obtained by numerical simulation to verify the accuracy of the model and the selected meso-parameters. Based on the numerical model, the effects of freeze–thaw damage on the mechanical properties of concrete were studied from the perspectives of the contact force between particles, crack development, and concrete cracking, and the characteristics of meso-scale failure of concrete model after freezing–thawing damage were analyzed. The results show that the stress–strain curves obtained by uniaxial compression simulation in the PFC3D-based numerical model of freezing–thawing concrete are consistent with the test results. During the uniaxial compression process in the freezing–thawing concrete model, the contact force at different stage points between the particles, crack development, and the number of cracked concrete fragments quantitatively reflected the degree of meso-scale concrete failure. With increasing numbers of freeze–thaw cycles, the failure severity of the model increased with obvious penetrating cracks, which is consistent with the failure pattern obtained by physical experiments. Therefore, numerical experiments can be used to study the meso-scale failure modes of concrete.

**Keywords:** C30 concrete; freezing–thawing damage; uniaxial compression test; particle flow; meso-scale failure

## 1. Introduction

Concrete is the dominant component of infrastructure around the world, for which the service environment has become increasingly complex. Freezing–thawing damage has become one of the main risk factors affecting concrete health and longevity, especially in cold areas [1,2]. Freezing–thawing cycles can affect the basic mechanical properties of concrete, such as compressive strength, tensile strength, and elastic modulus [3], further affecting the performance and load bearing capacity of concrete structures, which not only influence daily production and work, but also jeopardizes the safe operation and use of concrete structures.

Since the mid-20th century, freezing–thawing damage of concrete has been studied from the perspective of materials. Although some problems remain unresolved, a freezing–thawing damage

theory has been established [4–6], providing theoretical support for past and current research. Most of the studies on concrete performance during the freeze–thaw cycle are mainly focused on the degradation of mechanical properties of concrete. Specifically, during the concrete freezing–thawing damage process, micro-cracks and expansion appear due to the change in the internal pore structure of concrete, resulting in a change in mechanical properties such as the strength and deformation modulus of concrete [7]. At present, extensive experimental studies are carried out on the mechanical properties of concrete after freezing–thawing damage, but the test methods used have been quite different. Cao [8] studied the deterioration process of freezing–thawing concrete by uniaxial compression tests, and established the freezing–thawing damage equation of concrete. Hasan et al. [9] conducted uniaxial compression tests of concrete subjected to different freezing–thawing cycles, and studied the degradation law of the number of freezing–thawing cycles on the mechanical properties of concrete. Bogas et al. [10] performed uniaxial compression tests on concrete after freezing–thawing cycles, and clarified the relationships among dynamic elastic modulus, compressive strength, and freezing–thawing cycles. Shang et al. [11] completed biaxial compression tests on concrete cubes under different freezing–thawing cycles to study the effect of freezing–thawing cycle on the mechanical properties of concrete, and established the failure criteria and stress–strain relationship model of freezing–thawing concrete under biaxial compression. Ma et al. [12] compared the uniaxial compression constitutive model of concrete damaged by salt freezing with the corresponding uniaxial compression test results of salt-frozen concrete, and determined the influence of freeze–thaw cycles on the mechanical properties of concrete. Sun et al. [13] determined the damage parameters of concrete under the coupled conditions of salt solution and freezing–thawing cycles by putting concrete under freezing–thawing cycles in different salt solutions. Zhu et al. [14] established a finite element numerical model of uniaxial compression according to the change in numbers of freeze–thaw cycles, studied the influence of different freeze–thaw cycles on the mechanical properties of concrete, and compared it with the physical test results. The results of the numerical simulation and physical test were consistent. Hasan et al. [15] reported the stress–strain curve model of concrete under freezing–thawing cycles by fitting and analyzing the data obtained from a large number of freezing–thawing tests. Liu et al. [16] found the strength changing law of concrete through freezing–thawing tests of concrete, proposing a new freezing–thawing damage parameter to describe the process of concrete strength change, and introduced the stress damage equation of concrete. Duan et al. [17] added a new random damage variable for freezing–thawing degradation in concrete through freezing–thawing tests, and discussed the influence of this variable on the mechanical properties of freezing–thawing damage of concrete. Most of these experiments and numerical models focused on the macro-mechanical properties of freezing–thawing concrete, whereas the meso-scale characteristics of freezing–thawing concrete have not been studied in depth.

In this study, the effect of the number of freeze–thaw cycles on the thickness of the damaged layer and the uniaxial compressive strength of concrete were obtained using freezing–thawing damage testing. The particle flow discrete element method was used to establish the meso-scale mechanical model of freeze–thaw-damaged concrete specimens for uniaxial compression simulation. Based on the numerical model, the characteristics and modes of meso-scale failure of concrete freezing–thawing damage were analyzed, providing theoretical support for the control and prevention of freezing–thawing damage of concrete in cold regions.

## 2. Experimental Materials and Methods

### 2.1. Freezing–Thawing Tests of Concrete

In order to study the uniaxial compression failure morphology of concrete under different numbers of freezing–thawing cycles, a basis for meso-parameter calibration in a numerical simulation has been provided. The test specimen was C30 prismatic concrete, with dimensions of $100 \times 100 \times 300$ mm. The coarse aggregate was two graded pebbles with a roughly ellipsoid and oblate shape and density of 2700 kg/m$^3$. The fine aggregate was medium sand with fineness modulus of 2.5 and density of

2000 kg/m$^3$. The cement was po42.5 ordinary Portland cement with density of 3000 kg/m$^3$. There were 7 groups of prismatic concrete specimens, of which 6 were subjected to freezing–thawing cycles and the other group included the control specimens. Each group included three concrete specimens that were used to test the thickness of the concrete damage layer under different freezing–thawing cycles and the uniaxial compression strength after freezing–thawing cycles. The mix proportions of concrete specimens are listed in Table 1.

**Table 1.** Mix of C30 concrete.

| Specimen Strength | Water/Binder Ratio | Sand Ratio | 1 m$^3$ Material Content of Concrete/kg | | | | | |
|---|---|---|---|---|---|---|---|---|
| | | | Water | Cement | Sand | Stone | Fly Ash | Admixture |
| C30 | 0.53 | 0.4 | 170 | 256 | 765 | 1150 | 64 | 9.6 |

Four days before the freezing–thawing test, the specimens were removed from the maintenance site for appearance inspection, and then immersed in water at 18 °C for four days before the freezing–thawing test. Each freezing–thawing cycle lasted 4 h, in which freezing required 2.5 h and thawing required 1.5 h. The 6 different groups of specimens were frozen and thawed 25, 50, 75, 100, 125, or 150 times, while the control group specimens were maintained in the standard climate chamber. The freezing–thawing temperature of concrete was set to –22 °C. Once the freezing–thawing cycle was completed, uniaxial compression failure tests were carried out on all specimens at the same time.

### 2.2. Damaged Layer of Frozen-Thawed Concrete

Concrete specimens subjected to the various numbers of freeze–thaw cycles are shown in Figure 1. As concrete is categorized as a multiphase heterogeneous material, different degrees of micro-cracks appeared. As the freezing–thawing test proceeded, the number of micro-cracks on the surface of concrete increased after 25 freezing–thawing cycles, and the surface of the specimens began to be rough, with a small amount of cement mortar falling off. After 50 freezing–thawing cycles, cracks were visible on the surface of the specimens, which had become obviously rough; after 75 cycles, the micro-cracks on the surface of concrete gradually expanded, part of the surface began to peel off, and some aggregate was exposed. After 100 freezing–thawing cycles, the concrete surface seriously peeled off with some coming off in scales or layers, and aggregate was more exposed; after 125 freezing–thawing cycles, more concrete peeled off in scales or layers with some corners even peeling off. After 150 freezing–thawing cycles, aggregate was exposed in large areas and more areas in the corner peeled off.

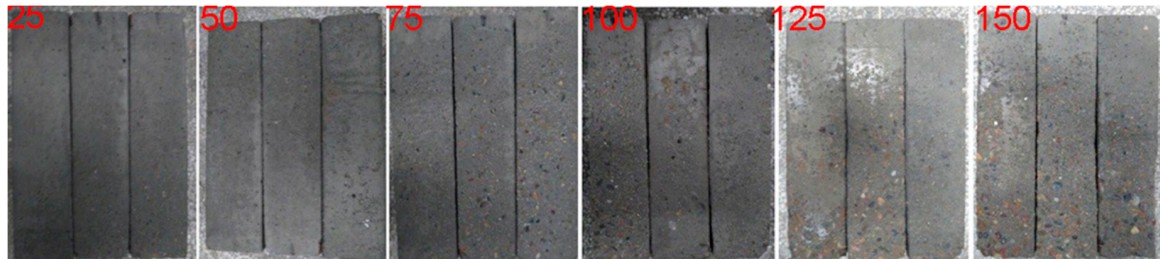

**Figure 1.** Concrete specimens with the different number of freezing–thawing cycles.

Concrete freezing–thawing is a gradual process. The outermost layer of concrete is seriously damaged, but the deeper in the concrete, the less the damage. Therefore, concrete is considered in three parts: freezing–thawing damage layer, freezing–thawing damage transition layer, and undamaged layer [18]. The freezing–thawing damage transition layer is a dense and thin area. To simplify the analysis of such behavior, this part of the concrete is considered the undamaged layer of concrete. The degradation of mechanical properties of freezing–thawing concrete is mainly caused by the degradation of mechanical properties of the damaged layer of concrete [19]. A simplified sketch of a

concrete section is shown in Figure 2a. This paper assumed that the damaged layer of concrete was uniformly spread around the section.

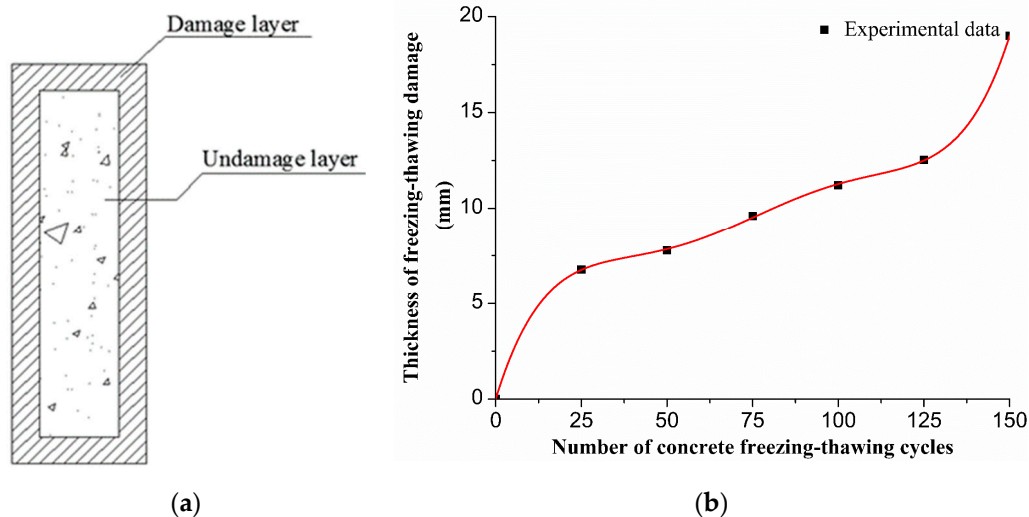

**Figure 2.** (**a**) Damaged layer distribution of freezing–thawing concrete section and (**b**) the curve of relationship between the number of freezing–thawing cycles and damage thickness.

The thicknesses of the damaged layer of the six groups of concrete specimens under the different numbers of freezing–thawing cycles were measured using ultrasonic nondestructive testing technology [20]. HC-U81 concrete strength defect detector was used to measure the thickness of concrete freeze–thaw damage layer, and the measurement accuracy was 0.01 mm. A concrete specimen was divided into three regions, in which the thicknesses of the damaged layer of concrete was detected by ultrasonic nondestructive testing, and the average value of the results was used. Using Origin software (OriginLab, USA), the polynomial fitting of the thickness $d$ of freezing–thawing damage layer with the number of freezing–thawing cycles $N$ was conducted, and the fitting curve is shown in Figure 2b.

Figure 2b shows that the thickness of the damaged layer of concrete increased with increasing number of freezing–thawing cycles, the growth rate of concrete damage layer thickness from 25 to 125 freeze–thaw cycles was basically unchanged, and the growth rate of the layer of damaged concrete thickness with more than 125 freeze–thaw cycles obviously accelerated. In the process of repeated freezing–thawing of free capillary water in concrete, the damage stress caused by volume expansion increased gradually, causing new micro-cracks to continue developing and damage to accumulate continuously. When the damage value exceeds a certain range, the damage will increase sharply.

*2.3. Uniaxial Compression Test of Concrete Damaged by Freezing–Thawing*

With the goal of determining the uniaxial compression failure mode and stress–strain curve of concrete damaged by freezing–thawing, 7 groups of specimens were tested under uniaxial compression on a microprocessor-controlled electro-hydraulic servo pressure tester. According to The Test Code for Hydraulic Concrete [21], the average value of peak stress of three specimens was taken as the compressive strength of the concrete specimens. During the test, the concrete specimen was preloaded at 2 KN, and then the lifting speed of the testing machine was set to 0.5 mm/s until the strain reached $3.0 \times 10^{-3}$, then the test stopped. The failure modes and stress–strain curves of freezing–thawing-damaged concrete under uniaxial compression were obtained as shown in Figures 3 and 4. The peak stress–strain curves are shown in Figure 5.

Figures 3 and 4 show that the severity of the uniaxial compression damage to concrete increased with the increase in the number of freezing–thawing cycles. When the concrete specimens without freezing–thawing were damaged, diagonal cracks mainly occurred. However, with the increase in

freezing–thawing cycles, concrete became more brittle and the risk of bursting became less serious. With 25 freezing–thawing cycles, the diagonal main cracks deepened and large breaks in flakes or bulk along the direction of the main cracks occurred. The situation became more obvious with the increase in the number of freezing–thawing cycles. So when the specimen was damaged, many vertical main cracks usually occurred. With increasing freezing–thawing cycles, the micro-cracks inside the concrete gradually increased [5], which further deformed the specimen transversely under compression, resulting in the vertical cracking of concrete for its transverse tension. With increasing the compressive load, the change law of stress–strain curve of freezing–thawing-damaged concrete under uniaxial compression was similar to the one shown above (Figure 4), with a linear elastic stage, pre-peak plastic stage, peak point, and post-peak decline stage. In the linear elastic stage, the elastic modulus decreased with the increase in the number of freezing–thawing cycles. The peak stress decreased with the increase in the number of freezing–thawing cycles while the peak strain increased with the increase in the number of freezing–thawing cycles. This occurred because the internal micro-structure of concrete specimens was damaged after freezing–thawing cycles. With the increase in the number of freezing–thawing cycles, the internal damage of concrete developed continuously, leading to a change in the concrete constitutive behavior.

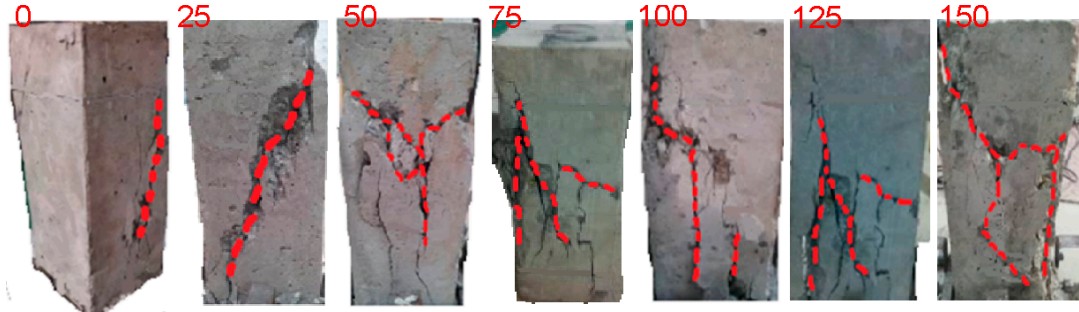

**Figure 3.** Failure modes of concrete under uniaxial compression with different numbers of freezing–thawing cycles.

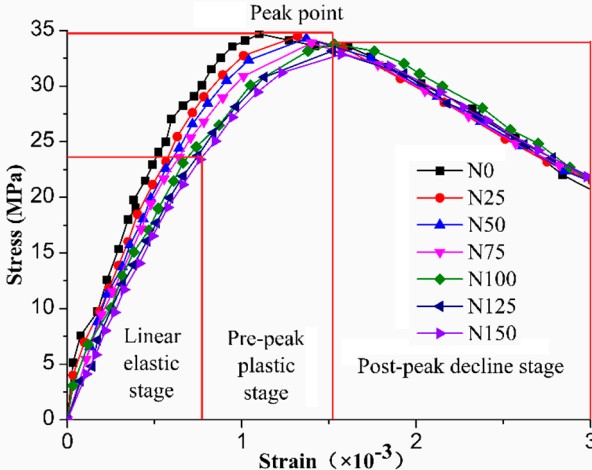

**Figure 4.** Stress–strain curve of concrete under different numbers of freezing–thawing cycles.

To study the change in the axial dimensions of concrete specimens before and after uniaxial compression under increasing number of freezing–thawing cycles, the axial height of specimens before and after compression was measured using Vernier calipers. Each specimen was measured three times in the axial direction, and the average value was used as the axial compression deformation of specimens. In each group, we measured three different specimens to determine the size change of

concrete specimens after uniaxial compression under different freezing–thawing cycles, as shown in Table 2.

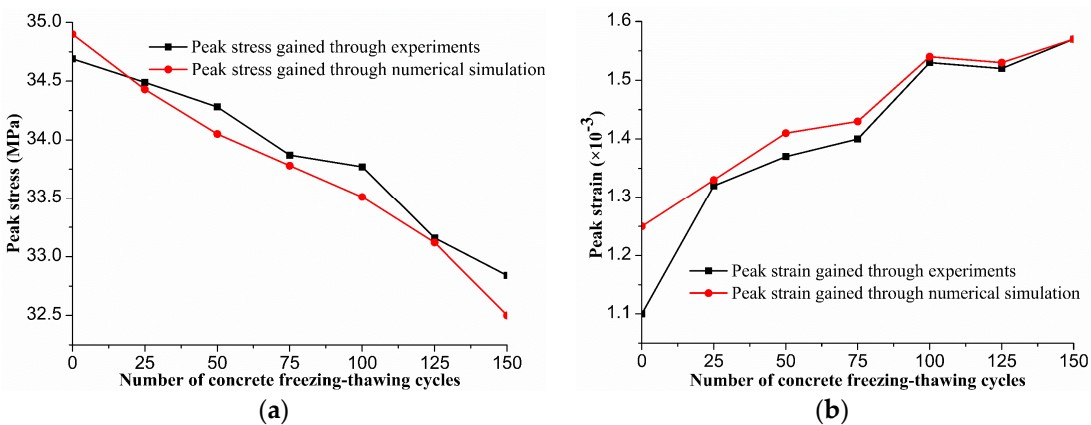

(a)  (b)

**Figure 5.** (**a**) Peak stress and (**b**) peak strain of concrete under different numbers of freezing–thawing cycles.

**Table 2.** Axial deformation of concrete specimens under different freeze–thaw cycles.

| Number of Freezing-Thawing Cycles | 0 | 25 | 50 | 75 | 100 | 125 | 150 |
|---|---|---|---|---|---|---|---|
| Average Value of Test Axial Deformation (mm) | 0.897 | 0.894 | 0.894 | 0.891 | 0.890 | 0.890 | 0.889 |

## 3. Numerical Model of Concrete Damaged by Freezing–Thawing

### 3.1. Model Setup

The basic principle of the discrete element method [22–24] is to treat the whole medium as a series of discrete and independent moving elements that have certain geometric (shape, size, arrangement, etc.), physical, and chemical characteristics. These elements are microscopic and only interact with adjacent elements, whose motion is controlled by the classical equations of motion. The deformation and evolution of the whole medium are described by the motion and mutual position of each element. Particle Flow Code (PFC) [25] is software based on the discrete element code, which allows discrete particles to undergo displacement and rotation. With the automatic identification of new contacts in the calculation process, particles can be combined to simulate deformed polyhedral particles [26,27]. In PFC software, the linear parallel bond model is usually used to simulate the contact between aggregate particles. The linear parallel bond model is regarded as many parallel springs with certain stiffness distributed on the contact surface of particles, which can transfer the force and moment between particles. When the linear parallel bond model is adopted, the total contact force $\overline{F}_i$. and total contact moment $\overline{M}_i$ between particles need to be calculated, as follows:

$$\begin{aligned} \overline{F}_i &= \overline{F}_i^n + \overline{F}_i^s \\ \overline{M}_i &= \overline{M}_i^n + \overline{M}_i^s \end{aligned} \tag{1}$$

where $\overline{F}_i^n$ is the normal contact force between particles, $\overline{M}_i^n$ is the normal contact moment between particles. $\overline{F}_i^s$ is the tangential contact force between particles, and $\overline{M}_i^s$ is the tangential contact moment between particles. The principle of the linear parallel bond model between particles is shown in Figure 6.

Generation of concrete numerical model specimens involves first generating the boundary wall in the calculation range, then generating the small spherical particles randomly in the boundary wall

range to simulate the cement mortar to fill the specimen, and using clump technology to generate the pebble coarse aggregate of different shapes, in which the spherical particles and the coarse aggregate are subject to a Gaussian distribution. A test piece of pebble concrete with good contact and compactness was generated; the coarse aggregate of the pebble included two kinds of coarse aggregates with oblate sphere and ellipsoid shapes, as shown in Figure 7a,b. The model size of the specimens was $100 \times 100 \times 300$ mm. Considering the influence of the mix ratio of concrete (Table 1) and the size of the cement mortar with the running speed of computer, we assumed that the cement mortar was composed of particles with radii of 2–5 mm, and the density was set to 2000 kg/m$^3$; the size of the coarse aggregate was the same as in the physical test, which was composed of 5–20 mm clump particles, and the density was set to 2700 kg/m$^3$. The cement mortar particles accounted for 51% of the total volume of particles and the coarse aggregate accounted for 49%. The total number of particles was 6384. The model was built with the Z axis as the vertical direction and the X and Y axes as the horizontal directions, as shown in Figure 7c. According to the thickness of freezing–thawing damage layers under different numbers of freezing–thawing cycles obtained in the previous test, the damaged and undamaged areas of concrete were established using PFC3D software, as shown in Figure 7d,e, in which the green area is the damaged area of frozen-thawed concrete, and the blue area is the undamaged concrete.

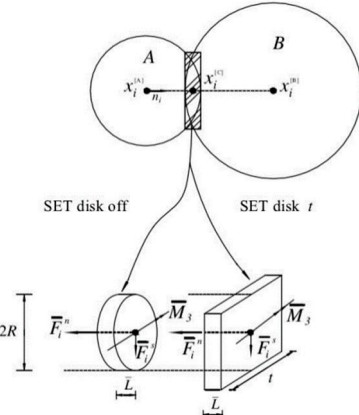

**Figure 6.** Schematic diagram of linear parallel bond model between particles.

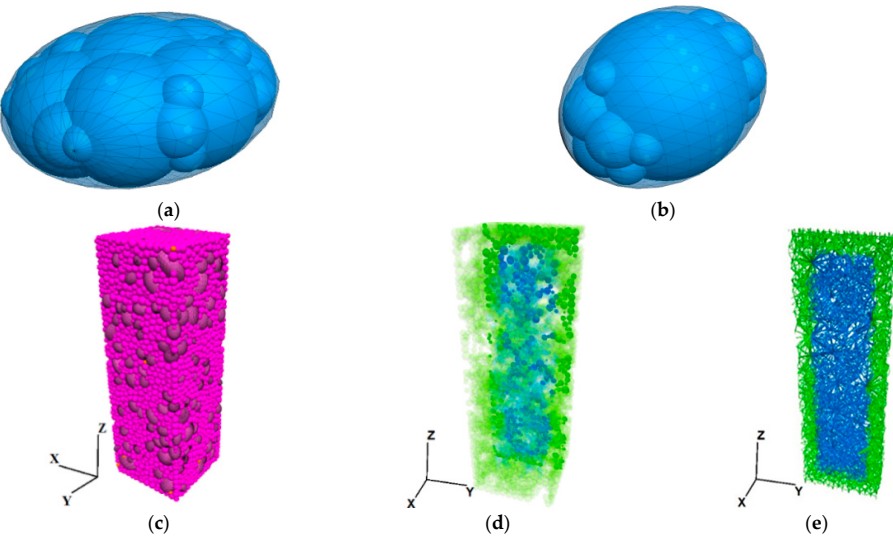

**Figure 7.** Model regional grouping diagram and numerical model of freezing–thawing cycle. (**a**) Oblate sphere coarse aggregate; (**b**) Ellipsoid coarse aggregate; (**c**) Freezing–thawing concrete model; (**d**) Regional grouping diagram of specimens model; € Grouping diagram of contact attribute area of specimen.

## 3.2. Calibration of Meso-Parameters

When the particle discrete element method was used for this concrete research, the biggest problem was the calibration of meso-parameters of the materials. Because the meso-parameters are needed for calculation in PFC, they cannot be obtained directly through experiments. Therefore, it was necessary to determine the meso-parameters suitable for macro-physical parameters through multiple numerical simulation tests after continuous selection and trial calculation to establish the relationship between them. Based on the various sensitivities of different meso-parameters to the macro-mechanical properties of concrete [28,29], in this study, the meso-parameters of the damaged area were calibrated by an inversion trial-and-error method according to the thickness of the freezing–thawing damage layer and stress–strain curve under uniaxial compression. The PFC numerical model generally only considers the meso-parameters between cement and mortar [30,31], but does not consider the meso-parameters of the weak interface between cement mortar and coarse aggregate. In this study, the meso-parameters were divided into two parts: the meso-parameters between the cement and mortar and the meso-parameters of the weak interface between cement mortar and coarse aggregate, as shown in Tables 3 and 4.

**Table 3.** Meso-parameters of cement mortar bonding of the damaged layer.

| Number of Freezing–Thawing Cycles | emod/Gpa | kratio | fric | pb_emod/Gpa | pb_kratio | pb_nstrength/Mpa | pb_strength/Mpa | pb_radius |
|---|---|---|---|---|---|---|---|---|
| 0 | 40 | 0.1 | 0.6 | 40 | 0.42 | 68 | 68 | 0.5 |
| 25 | 32 | 0.1 | 0.6 | 32 | 0.42 | 64 | 64 | 0.5 |
| 50 | 30 | 0.1 | 0.6 | 30 | 0.42 | 62.25 | 62.25 | 0.5 |
| 75 | 28 | 0.1 | 0.6 | 28 | 0.42 | 61.6 | 61.6 | 0.5 |
| 100 | 26 | 0.1 | 0.6 | 26 | 0.42 | 63 | 63 | 0.5 |
| 125 | 24.5 | 0.1 | 0.6 | 24.5 | 0.42 | 61 | 61 | 0.5 |
| 150 | 26 | 0.1 | 0.6 | 26 | 0.42 | 62 | 62 | 0.5 |

**Table 4.** Meso-parameters of cement aggregate bonding of the damaged layer.

| Number of Freezing–Thawing Cycles | emod/Gpa | kratio | fric | pb_emod/Gpa | pb_kratio | pb_nstrength /Mpa | pb_strength /Mpa | pb_radius |
|---|---|---|---|---|---|---|---|---|
| 0 | 40 | 0.1 | 0.6 | 40 | 0.42 | 34 | 34 | 0.5 |
| 25 | 32 | 0.1 | 0.6 | 32 | 0.42 | 32 | 32 | 0.5 |
| 50 | 30 | 0.1 | 0.6 | 30 | 0.42 | 31.25 | 31.25 | 0.5 |
| 75 | 28 | 0.1 | 0.6 | 28 | 0.42 | 30.8 | 30.8 | 0.5 |
| 100 | 26 | 0.1 | 0.6 | 26 | 0.42 | 31.5 | 31.5 | 0.5 |
| 125 | 24.5 | 0.1 | 0.6 | 24.5 | 0.42 | 30.5 | 30.5 | 0.5 |
| 150 | 26 | 0.1 | 0.6 | 26 | 0.42 | 31 | 31 | 0.5 |

Note: emod represents the effective modulus of the linear part in the meso-model of concrete, kratio represents the normal to tangential stiffness ratio of the linear part, fric represents the friction coefficient between particles, pb_emod represents the effective modulus of the parallel bonding part, pb_kratio represents the normal to tangential stiffness ratio of the parallel bonding part, pb_nstrength represents the normal strength of the parallel bonding part, pb_strength represents the tangential strength of the parallel bonding part, and pb_radius represents the parallel bond radius.

## 3.3. Model Validation

With the discrete element software PFC3D, the uniaxial compression of freezing–thawing concrete was simulated using the meso-parameters of freezing–thawing concrete in Tables 3 and 4, keeping the upper wall still, and applying an upward velocity of 0.5 mm/s to the lower wall to simulate the upward lifting of the testing machine to obtain the peak stress and peak strain of the concrete specimen model under different freezing–thawing cycles, as shown in Figure 5. The stress–strain curve of concrete obtained by numerical simulation is shown in Figure 8.

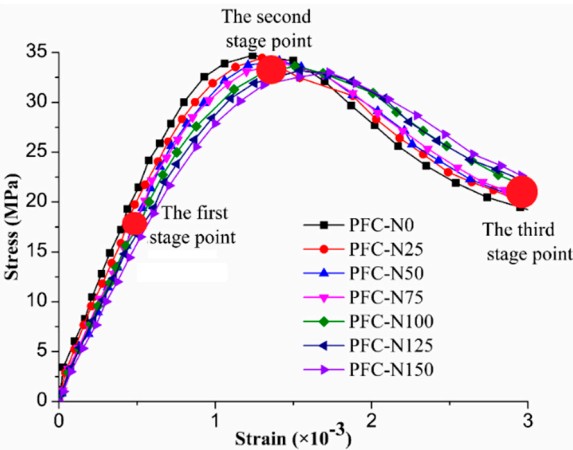

**Figure 8.** Stress–strain curve of concrete under freezing–thawing cycles by Particle Flow Code (PFC) stimulation.

Figure 4, Figure 5, and Figure 8 show that the freezing–thawing damage curve of concrete simulated by PFC3D was basically consistent with the law obtained by experiment in the rising section and stress–strain peak value. However, in the post-peak decline stage after the peak stress was reached, the bonds between particles compressed and fractured, the restriction and binding force of particles decreased, the shear resistance of materials decreased, and the deformation of the model changed to the state of shear expansion. Because the particle shape in the numerical model is single and the bite force is weak, the post-peak softening of the numerical model is enhanced. When there were no or few freezing–thawing cycles, the curve simulated by PFC decreased faster; the higher the number of freezing–thawing cycles, the slower the curve decreased. When the stress exceeded the peak stress, the softening effect of concrete slowed with increasing numbers of freezing–thawing cycles. In general, the stress–strain curves of concrete obtained by PFC3D simulation were in good agreement with the experimental data, which shows that the model established in this paper can be used to simulate the mechanical properties of concrete after freezing–thawing damage under uniaxial compression.

The axial displacement of concrete model particles can be used to reflect the compression deformation of model specimens. Therefore, the accuracy of the concrete numerical model and the meso-parameters was verified by comparing the axial displacement of the concrete numerical model at the strain of $3 \times 10^{-3}$ with the axial compression deformation of the concrete specimen. Table 5 shows the comparison between the test axial compression deformation and the particle axial displacement.

**Table 5.** Comparison of uniaxial compression deformation and particle axial displacement of concrete under different numbers of freezing–thawing cycles.

| Number of Freezing–Thawing Cycles | Average Value of Test Axial Deformation (mm) | Axial Displacement of Particles (mm) | Difference Value between Numerical Model and Test in Axial Deformation (mm) | Relative Error of Numerical Model and Test in Axial Deformation (%) |
|---|---|---|---|---|
| 0 | 0.897 | 0.899 | 0.002 | 0.223 |
| 25 | 0.894 | 0.895 | 0.001 | 0.112 |
| 50 | 0.894 | 0.898 | 0.004 | 0.447 |
| 75 | 0.891 | 0.894 | 0.003 | 0.337 |
| 100 | 0.890 | 0.896 | 0.006 | 0.674 |
| 125 | 0.890 | 0.897 | 0.007 | 0.787 |
| 150 | 0.889 | 0.890 | 0.001 | 0.112 |

Table 5 shows that with the increase in freezing–thawing cycles, the axial deformation obtained by the test gradually reduced, and the axial displacement of particles was not much different. This occurred because with the increase in freezing–thawing cycles, the damage degree of concrete increased, and the height of concrete specimen decreased gradually. When reaching the same strain of $3 \times 10^{-3}$, the

compression deformation decreased, while the height of the numerical simulation specimen did not change before uniaxial compression, so the axial displacement of particles differed little. The axial deformation obtained from the test was very close to the axial displacement of particles obtained from the numerical simulation, and the relative error was not more than 1%, which showed that the concrete numerical model and the calibrated meso-parameters were correct. The axial displacements of particles in the 75 and 150 cycles of freezing–thawing concrete numerical model are shown in Figure 9.

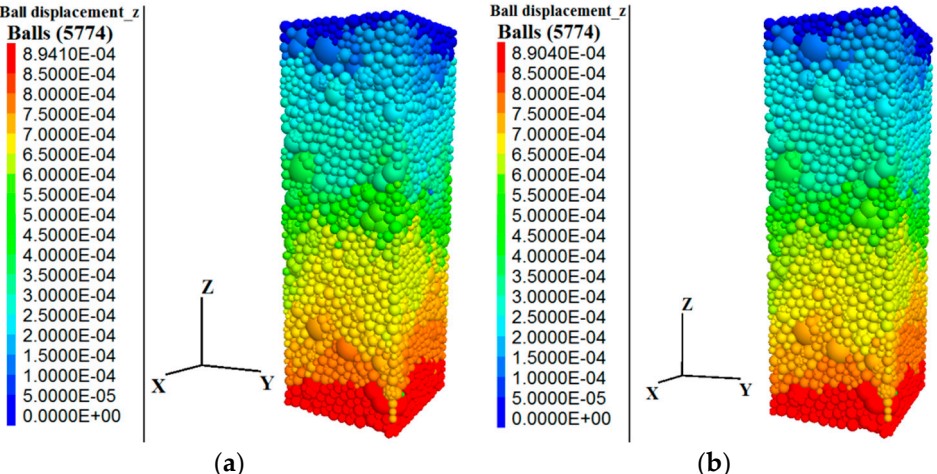

**Figure 9.** Axial displacement of concrete particles under (**a**) 75 and (**b**) 150 freezing–thawing cycles.

Figure 9 shows that the axial displacement of particles after 75 cycles of freezing and thawing was 0.894 mm, and that of particles after 150 cycles of freezing and thawing was 0.890 mm. In the numerical simulation, the bottom wall was lifted upward to simulate the upward lifting process of pressure testing machine in the macro physical test, so the displacement of the upper particles of concrete was 0, and the axial displacement was the displacement of the lower particles of concrete.

## 4. Numerical Simulation of Meso-Scale Failure of Freezing–Thawing Damage Concrete under Uniaxial Compression

Figures 4 and 8 show that the uniaxial compression process of concrete could be divided into three stage points: the first stage point was when the stress of the specimens in the curve rising section reached 0.5 times the peak stress, the second stage point was when the stress of the concrete specimens reached the peak stress, and the third stage point was when the strain of the specimens reached $3.0 \times 10^{-3}$. Because the concrete became damaged after the second stage point in the uniaxial compression process, the second and third stage points were selected for analysis and research in this paper. Based on the PFC3D numerical model, the effects of freezing–thawing damage on the mechanical properties of concrete were studied from the aspect of contact force between particles and the development of cracks. The cracking of specimens and the meso-scale failure mode of freezing–thawing concrete were analyzed.

### 4.1. Analysis of Contact Force between Concrete Particles

In the PFC model, the contact force between particles is the force produced by extrusion and shear between particles. The contact force between particles can quantitatively reflect the damage degree of concrete under uniaxial compression. Therefore, we studied the maximum value of the contact force between particles in the two different stage points of freezing–thawing-damaged concrete, and analyzed the change in contact force between particles during uniaxial compression of the concrete model under different numbers of freezing–thawing cycles. The maximum contact force at the two different stage points is shown in Figure 10.

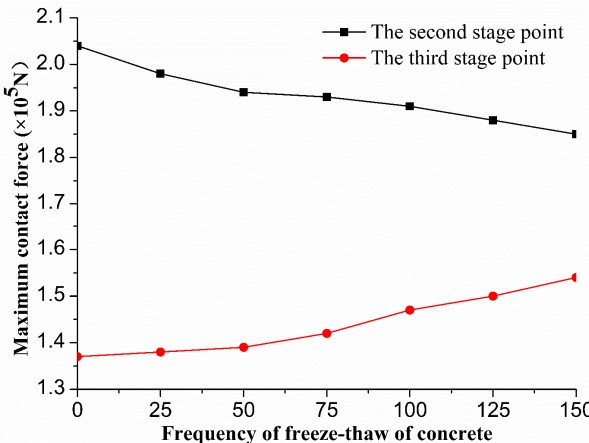

**Figure 10.** Maximum contact force of freezing–thawing-damage concrete.

Figure 10 shows that with increasing freezing–thawing cycles, the maximum contact force between particles in the second stage point gradually decreased, and the maximum contact force between particles in the third stage point gradually increased; the maximum contact force at the second stage point was significantly higher than the maximum contact force between particles in the third stage point because with the increase in freezing–thawing cycles, at the second stage point, the bite force between particles gradually decreased, and the peak stress gradually decreased, so the maximum contact force between particles gradually decreased. With the increase in freezing–thawing cycles, the residual stress of the model gradually increased, so the maximum contact force between particles in the third stage point gradually increased, but the residual stress of the model was obviously less than the peak stress, so the maximum contact force in the second stage point was obviously higher than the maximum contact force in the third stage point. Contact force in the second stage point with 75 and 150 cycles of freezing–thawing is shown in Figure 11; the third stage point contact force is shown in Figure 12.

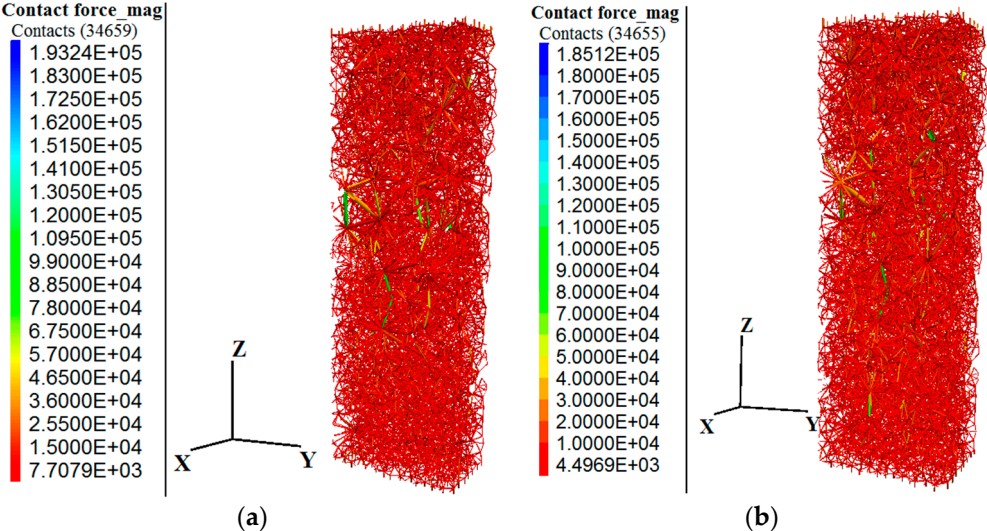

**Figure 11.** Contact force distribution of concrete during the second stage point (unit: N) with (**a**) 75 and (**b**) 150 cycles of freeze–thaw.

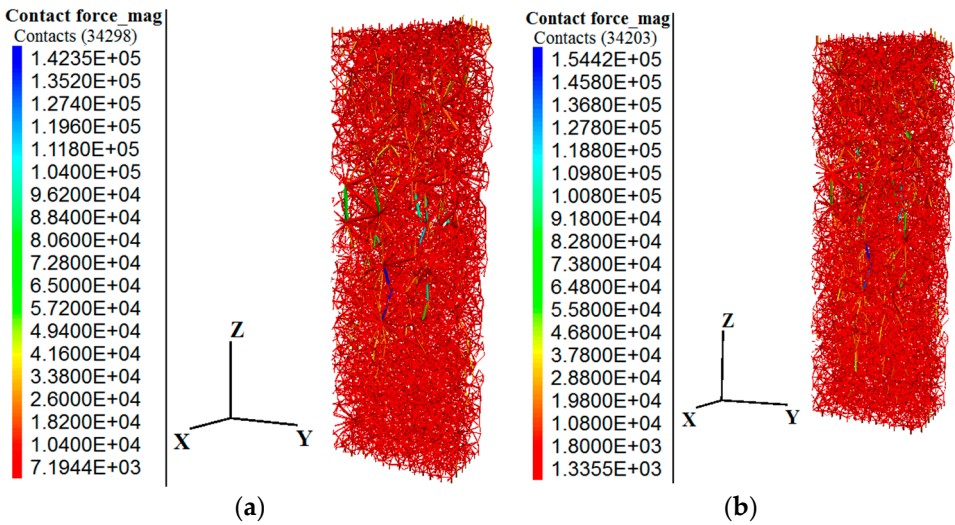

**Figure 12.** Contact force distribution of concrete at the third stage point (unit: N) with (**a**) 75 and (**b**) 150 cycles of freeze–thaw.

Figures 11 and 12 show that the maximum contact force between particles with 75 freeze–thaw cycles was 193.24 KN, and that between particles with 150 cycles was 185.12 KN at the second stage point. The maximum contact forces between particles with 75 and 150 cycles were 142.35 and 154.42 KN at the third stage point, respectively. In the second stage point, the maximum contact force between particles with 150 cycles of freezing–thawing was 4% lower than that of 75 cycles, which is basically consistent with the peak stress reduction rate of concrete under the same freezing–thawing cycles. In the third stage point, the maximum contact force between particles with 150 cycles of freeze–thaw increased by 7.8% compared with 75 cycles, which is basically consistent with the residual stress growth rate of the concrete model under the same freezing–thawing cycles under uniaxial compression strain of $3.0 \times 10^{-3}$. This shows that the modeled contact force of freezing–thawing concrete particles is correct. By controlling the termination condition of uniaxial compression of concrete numerical model, the contact force between particles at any point on the stress–strain curve is obtained to estimate the damage degree of the concrete structure at different points.

*4.2. Crack Development Law of Concrete Model*

In the concrete model, the contact between particles is a linear parallel bond (Pb). In the Pb model, when the stress value of the Pb exceeds the value of the Pb strength, the Pb frame breaks. At this time, a micro-crack occurs; when the shear bond strength value is exceeded, the generated crack is a shear crack; when the normal bond strength value is exceeded, the generated crack is a tension crack. For the Pb model, once the bond between particles breaks, the internal stress of concrete is redistributed, and the bond between adjacent particles also breaks. Then, through the connection and expansion of micro-cracks, the concrete material shows macro damage. Therefore, we studied the number of cracks after uniaxial compression at two different stage points of freezing–thawing-damaged concrete, and analyzed the micro-crack state of two different stage points of the freezing–thawing-damaged concrete model. The number of cracks in the two different stage points of concrete uniaxial compression is shown in Figure 13.

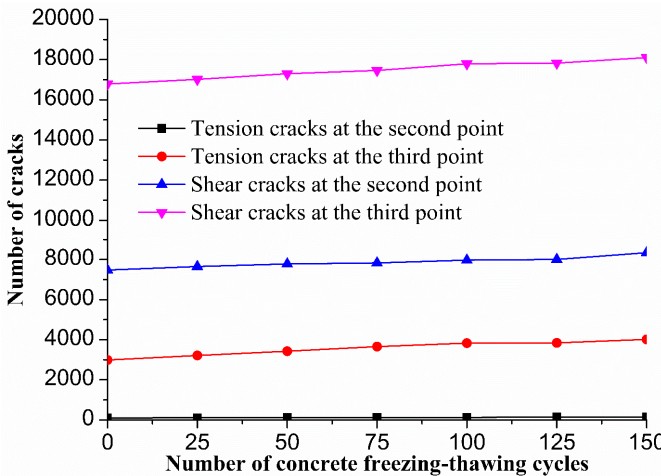

**Figure 13.** The number of cracks in two stage points of freezing–thawing-damaged concrete.

Figure 13 shows that with the increase in freezing–thawing cycles, the number of cracks in the concrete model gradually increased. The number of cracks in the second stage point was significantly lower than in the third stage point, and the number of shear cracks in the same stage point was significantly higher the number of tension cracks. This occurred because with the increase in freezing–thawing cycles, the internal damage of concrete was more serious and the number of micro-cracks increased. The second stage point was the peak stress point of concrete. At this time, the concrete specimen began to lose load capacity. At the third stage point, the concrete was in the post-peak softening stage. At this time, the concrete lost bearing capacity completely, so the number of micro-cracks increased significantly. Due to the large shear force between particles in the process of uniaxial compression, the number of shear cracks at the same stage point was significantly higher than the number of tension cracks. The number of concrete cracks in the second stage point of 75 and 150 freezing–thawing cycles is shown in Figure 14; the number of cracks in the third stage point is shown in Figure 15.

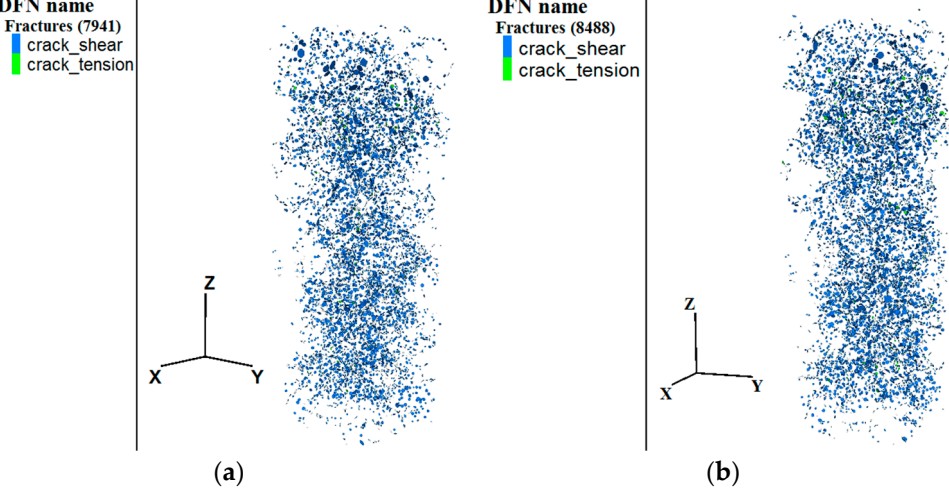

**Figure 14.** Number of concrete cracks at the second stage point with (**a**) 75 and (**b**) 150 freeze–thaw cycles.

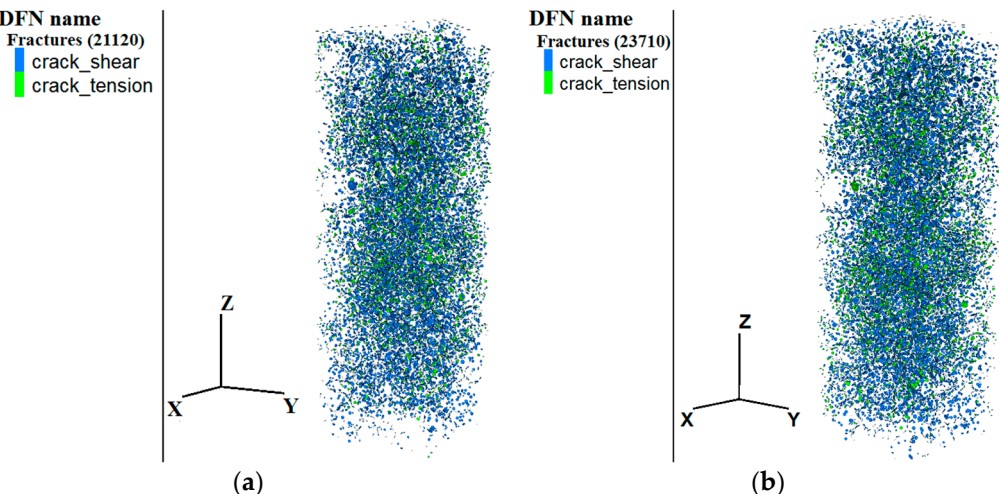

**Figure 15.** Number of concrete cracks at the third stage point with (**a**) 75 and (**b**) 150 freeze–thaw cycles.

Figures 14 and 15 show that at the second stage point, the number of cracks between particles with 75 freezing–thawing cycles was 7941, and the number of cracks between particles with 150 freezing–thawing cycles was 8488. In the third stage point, the number of cracks between particles 75 freezing–thawing cycles was 21,120, and the number of cracks between particles with 150 freezing–thawing cycles was 23,710. In the second stage point, the crack growth rate for 75 cycles was 6.8% that of 150 freezing–thawing cycles; in the third stage point, the crack growth rate for 75 cycles was 12.2% that of 150 cycles of freezing–thawing. Therefore, the crack growth rate in the third stage point was significantly higher than in the second stage point, and the cracks generated in the uniaxial compression of freezing-thawing concrete mainly occurred in the post-peak decline stage. By controlling the termination condition of uniaxial compression of the concrete numerical model, the number of cracks at any point on the stress–strain curve is obtained to predict the damage degree of concrete structure at different points.

*4.3. Cracking Law of Concrete Model Specimens*

When the micro-cracks in the concrete model connected and expanded, the concrete model showed a macro failure state, which is represented by the number of fragments in PFC3D software. The fragment was generated by the penetration of many micro-cracks around the particles. The larger the number of fragments, the more micro-cracks around the particles. When the concrete stress reached the peak stress, the crack of the specimen was just beginning, the crack was in the initiation stage, and the number of fragments was 0. Therefore, we only displayed the crack situation at the third stage point, as shown in Figure 16.

Figure 16 shows that the growth rate of the fragment in the third stage point was basically the same as that of the damaged layer because with the increase in freezing–thawing cycles, the thickness of the damaged layer gradually increased. The more serious the internal damage, the more the fragments are produced by uniaxial compression, so the growth rate was basically the same. Simultaneously, with the increase in freezing–thawing cycles, the number of concrete fragments under uniaxial compression gradually increased and the damage worsened. This occurred because with the increase in freezing–thawing cycles, the internal damage of the concrete continued to develop, and the number of micro-cracks produced by uniaxial compression gradually increased, which led to the connection and expansion of micro-cracks and the formation of more fragments, increasing the seriousness of damage, which is basically consistent with the compression failure pattern obtained from the test. The particle fragment at the third stage point is shown in Figure 17.

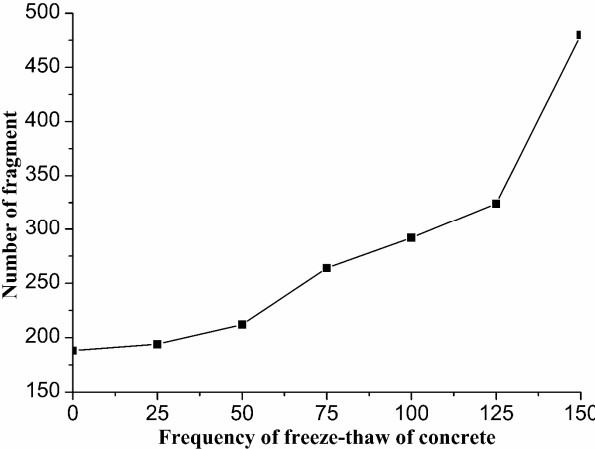

**Figure 16.** Number of fragments at the third stage point of freezing–thawing concrete.

Figure 17 shows that the maximum number of fragments damaged by concrete particles that did not undergo freeze–thaw was 188; the maximum numbers of particle fragments damaged by 25, 50, 75, 100, 125, and 150 cycles of freezing–thawing were 194, 212, 264, 292, 324, and 480, respectively. With the increase in freeze–thaw cycles, the failure mode of concrete was increasingly severe. The number of concrete fragments and the final failure morphology were used to predict the failure mode and the damage degree of concrete in the macroscopic physical test.

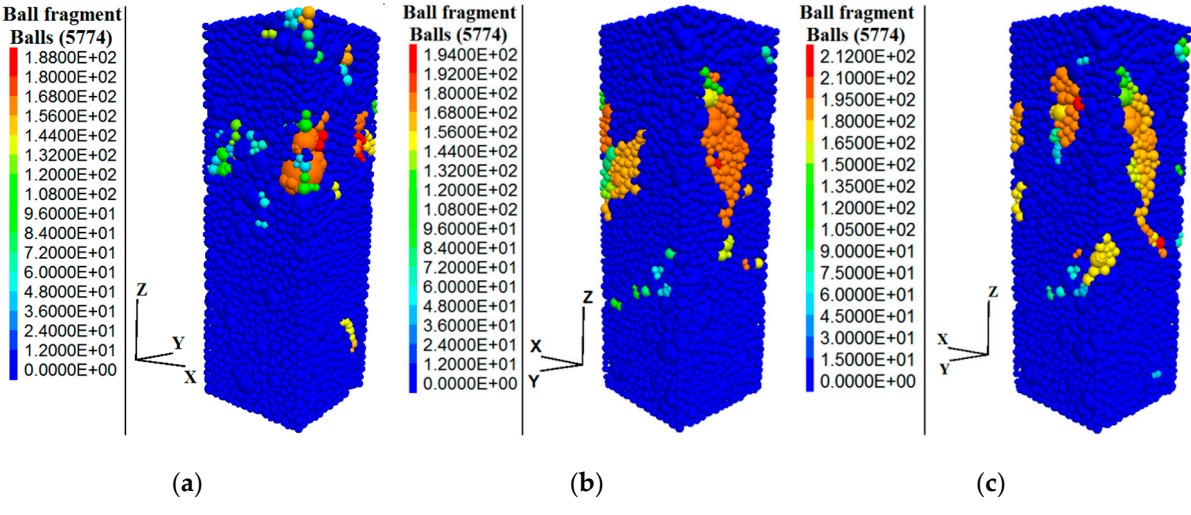

(a)　　　　　　　　　　(b)　　　　　　　　　　(c)

**Figure 17.** *Cont.*

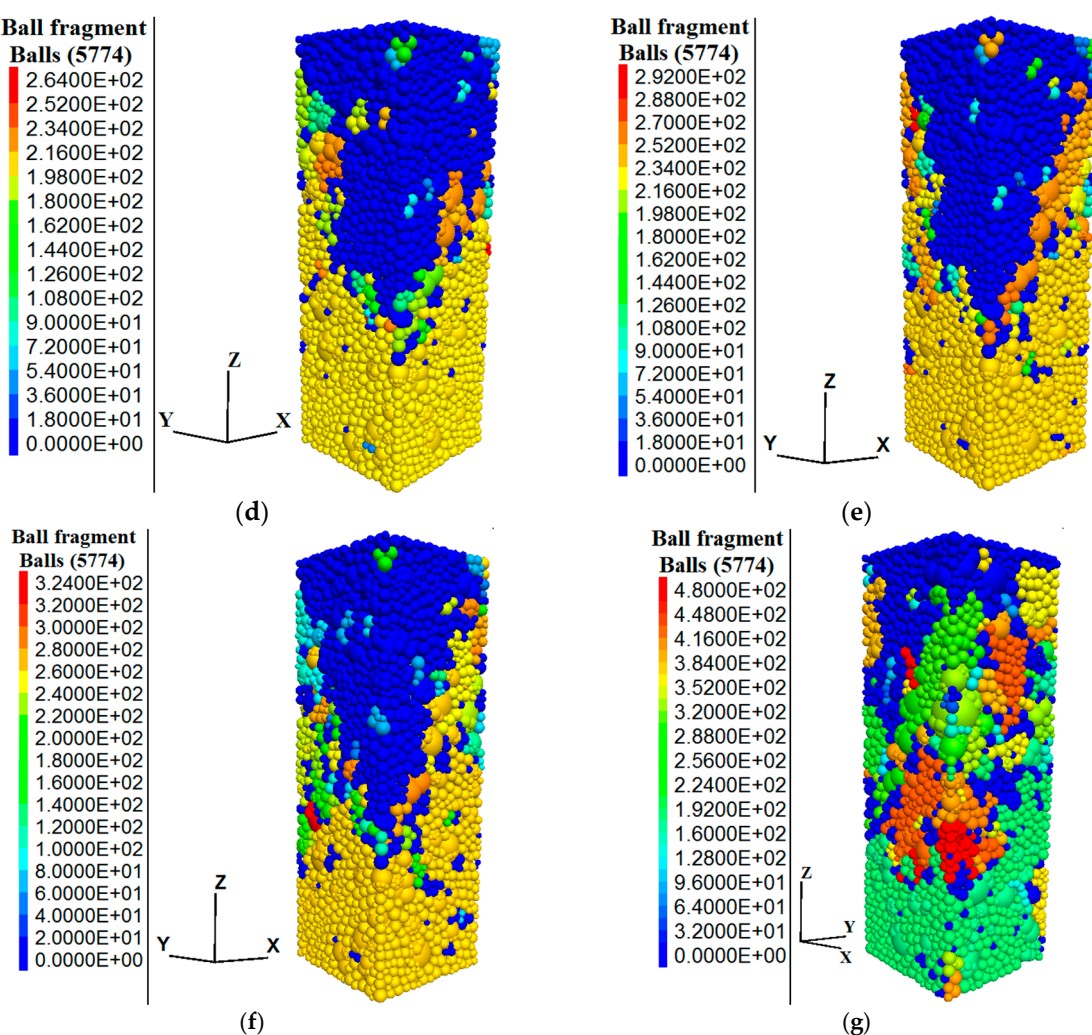

**Figure 17.** Failure states of concrete at the third stage point under different freezing–thawing cycles. (**a**) Failure state concrete without freeze–thaw; (**b**) Failure state of 25 freezing–thawing concrete; (**c**) Failure state of 50 freezing–thawing concrete; (**d**) Failure state of 75 freezing–thawing concrete; (**e**) Failure state of 100 freezing–thawing concrete; (**f**) Failure state of 125 freezing–thawing concrete; (**g**) Failure state of 150 freezing–thawing concrete.

## 5. Conclusions

Based on the uniaxial compression test of freezing–thawing-damaged concrete, a numerical model was established using particle flow discrete element theory to study the influence of freezing–thawing damage on the meso-scale failure mode of concrete. The following conclusions are drawn.

As the difference between the uniaxial compression axial deformation of the freezing–thawing concrete model and that of the numerical simulation was less than 1%, the model of freezing–thawing concrete and the selected parameters is accurate.

Through analysis of the contact force between particles in two different stage points of the numerical model, we found that the softening effect of concrete after the peak force is considerably weakened by the freezing–thawing effect on concrete. From 75 to 150 cycles of freezing–thawing, the softening effect after peak decreased by 39.7%. At the same time, by controlling the termination condition of uniaxial compression of the concrete numerical model, the contact force between particles at any point on the stress–strain curve can be used to estimate the damage degree of the concrete structure at different points.

With the increase in freezing–thawing cycles, the number of cracks increased significantly. In the second stage point, the crack growth rate from 75 to 150 cycles of freezing–thawing was 6.8%; in the third stage point, the crack growth rate from 75 to 150 cycles of freezing–thawing was 12.2%. In the third stage point, the crack growth rate was significantly higher than that in the second stage point, which indicated that the failure of concrete mainly occurred in the post-peak softening stage. By controlling the termination condition of uniaxial compression of the numerical model of concrete, the number of cracks at any point on the stress–strain curve is obtained to predict the degree of concrete damage.

With the increase in freezing–thawing cycles, the number of concrete fragments increased gradually, and the damage worsened. From 125 to 150 cycles of freezing–thawing, the growth rate of concrete fragments was the highest, with an increase of 48.1; the fragmentation after 150 cycles of freezing–thawing was the most serious. Therefore, testing the mechanical properties of concrete is necessary when concrete has undergone more than 125 cycles of freezing and thawing. At the same time, the number of concrete fragments and the final failure morphology are used to predict the failure mode and the damage degree of concrete in macroscopic physical test.

**Author Contributions:** Conceptualization, Z.S.; Data curation, X.D.; Formal analysis, Z.S. and X.D.; Investigation, X.D. and L.H.; Project administration, L.H. and Y.L.; Resources, Z.S., L.H. and Y.L.; Validation, Y.L. All authors have read and agreed to the published version of the manuscript.

**Funding:** This study was financially supported by the National Natural Science Foundation of China (Nos. 51879217, 51409207, and 51609200) and the National Science Foundation for Post-Doctoral Scientists of China (No. 2015M582765XB).

**Acknowledgments:** The author would like to thank the reviewers for their comments.

**Conflicts of Interest:** The authors declare no conflict of interest.

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
