# Peer review of "Meso-Scale Failure of Freezing–Thawing Damage of Concrete under Uniaxial Compression"

_applsci, doi:10.3390/app10041252_

Round 1

Reviewer 1 Report

The considered topic is suitable for investigation; the reported results can be a useful object for further analysis. The submission can be considered for the publication only after major modification. The main aspects required an improvement are the following:

1) The writing style is below acceptable limits. The text contains numerous grammatical errors and inaccuracies; the authors used the wrong terminology. The manuscript must be entirely rewritten, improving the language. Sometimes, the reviewer was unable to understand explanations. Due to an excessive number of such occurrences, only several corrections are highlighted in the attached manuscript.

2) Most of the results are presented without any comments. That is not acceptable for scientific work.

3) Conclusions are not informative and must be rewritten.

The attached manuscript contains more detail description of the required modifications (highlighted in the comments).

Reviewer 2 Report

The paper is very interesting and the results well presented.

In my opinion, the paper can be accepted, but I suggest to review the language and to better describe the phisical meaning of the input parameters used in the PFC3D environment. Also, I suggest to specify the temperature considered for the Freezing-thawing  

Author Response

Response to Reviewer 2 Comments

Point 1:In my opinion, the paper can be accepted, but I suggest to review the language and to better describe the phisical meaning of the input parameters used in the PFC3D environment. Also, I suggest to specify the temperature considered for the Freezing-thawing.

Response 1: I have retouched the language of the article and revised the grammar and inaccuracies. The input parameters of PFC software are meso parameters, which have no practical physical significance, and can only be obtained through macro physical test. It has been added that the freezing-thawing temperature of concrete is - 22 ℃.

Reviewer 3 Report

Please see comments in the pdf file.

Author Response

Response to Reviewer 3 Comments

Point 1:What does Sand Ratio mean ?

Response 1: Sand Ratio=Amount of sand/(Amount of sand+Amount of stone)

Point 2:What is the age of concrete at the time of testing?

Response 2:The age of concrete used in the test is 28 days

Point 3:should explain how the data fitting (Figure 2 (b)) is done.

Response 3:The data fitting in Figure 2 (b) has been explained.

Point 4:Figure 4 is this the average of each group of specimens? How is the variance for each group? The difference of peak stress between N0 and N150 is only 3 MPa, while the difference of three tests of one group could be larger.

Response 4:Figure 4 is not the average value of each group of test pieces, and the difference between each group of test pieces is not more than 15%.The average value of peak stress of three specimens is taken as the result of compressive strength of this group of specimens. When the difference between the maximum value of peak stress or one of the minimum values of peak stress and the intermediate value of peak stress in three concrete specimens is less than 15% of the intermediate value, the compressive strength of the intermediate value shall be taken; When the difference between the maximum value of peak stress and the minimum value of peak stress and the middle value of peak stress in three concrete specimens is greater than 15% of the middle value, this group of tests will be repeated. The stress-strain curve obtained by uniaxial compression of concrete specimens with the minimum difference between the concrete strength and the peak stress of three specimens is selected as the final stress-strain curve.

Point 5:How does this size of sand compare to the reality?

Response 5:The particle size of the cement mortar used in the simulation is 10 times larger than that of the sand used in the test.The size of sand used in the test is 0.35-0.5mm. Because the size of the model is certain, the smaller the size and the more the number of spherical particles, the larger the computer memory needed.Therefore, the particle size of cement mortar used in the model is increased by 10 times.

Point 6:The constitutive equations of the discrete model should be included.

Response 6:The discrete model used in this paper is linear parallel bond model, and the constitutive equation of linear parallel bond model has been listed in manuscript 3.1.

Point 7:Can the discrete model take into account the different particle distribution and provide also variance in the results as in reality?

Response 7:All the particle distribution in the discrete model is Gaussian distribution, which is basically consistent with the actual results.

Point 8:how is freezing-thawing calculated in the model? Aren't there thermal parameters which couple with the mechanical mode.

Response 8:The freezing-thawing of the model is expressed by different meso-parameters. The meso-parameters are different with different freezing-thawing times. There is no need for thermal parameters coupled to the mechanical model.

Point 9:How does the elastic modulus of each group compare to that of the experiment?

Response 9:The difference between the elastic modulus of each group and the experimental group is no more than 3%.

Point 10:How is the contact force determined in relation with the number of freezing-thawing cycles?

Response 10:The meso-parameters of the model are different with different freezing-thawing cycles. According to the PFC software, the contact force is different.

Point 11:How much of opening is considered a crack?

Response 11:The contact fracture between particles will form a tiny crack.

Point 12:Figure 16 seems the exact shape as the thickness of damage. Explanations or comments?

Response 12:The relationship between Figure 16 and the thickness of the damaged layer (Figure 2 (b)) has been explained in this paper.

Point 13:How does the adopted model compare to the Lattice Discrete Particle Model (LDPM) in terms of model setup and capabilities?

Response 13:PFC is simple to build concrete model and it can build complex models. At the same time, PFC model can be used to get the distribution of contact force and the number of cracks.

Point 14:What benefits does the research bring to practice? If the model can predict strength and modulus change with freezing-thawing, what/how many tests are required for calibrating the model; what are the limits of the model?

Response 14:This research can let researchers get the distribution of contact force and the number of micro cracks of concrete under the condition of known freezing-thawing times. It is a very long process to calibrate the meso-parameters. Generally, the sensitivity analysis of the meso-parameters is used to calibrate the parameters.

Point 15:The overall grammar and use of words need to be extensively improved.

Response 15:I have revised the points in the article, and revised the grammar and inaccuracies.

Round 2

Reviewer 1 Report

The writing style is still unacceptable. A native speaker should carefully revise entire text by improving it stylistically. This is a strict requirement for further consideration of this work. Please, re-submit the stylistically appropriate work – the reviewer will not spend any more time for improving the writing style. The modification must also address the drawbacks commented in the attached document.

Author Response

Response to Reviewer 1 Comments

Point 1:The writing style is still unacceptable. A native speaker should carefully revise entire text by improving it stylistically. This is a strict requirement for further consideration of this work. Please, re-submit the stylistically appropriate work – the reviewer will not spend any more time for improving the writing style. The modification must also address the drawbacks commented in the attached document.

Response 1:I have edited in English through MDPI service and revised it according to the opinions of reviewers.

Point 2:This wording is stylistically imperfect. The reviewer recommends to rephrase it as follows "extensive experimental studies are carried out..."

Response 2: I have made changes in the manuscript.

Point 3: This reference is wrong because of the multiple authors. The article must be cited as "Hasan et al. [9]..." Such errors are also found throughout the text. Please correct the references correspondingly.

Response 3:I have revised the references in the manuscript.

Point 4:The reviewer disagrees with this statement. There is proposed no expression to describe this relationship. The reviewer recommends rephrasing this statement as "...the effect of the number of freeze-thaw cycles on the thickness of the damaged layer and <...> is investigated."

Response 4:I have revised the manuscript according to the reviewer's opinion.

Point 5:Please describe (in the text) the technique used to determine the density of the aggregates.

Response 5:The density of aggregate is obtained by marking the name plate of aggregate when purchasing aggregate.

Point 6:Too many "maintenance room"... The reviewer suggests using the terms "climate chamber" or "laboratory conditions".

Response 6:The maintenance room has been changed to climate chamber.

Point 7:Please specify (in the text) the accuracy of this using ultrasonic.

Response 7:I have added in the text that HC-U81 concrete strength defect detector is used to measure the thickness of concrete freeze-thaw damage layer, the measurement accuracy is 0.01mm.

Point 8:Please change the structure of the sentence: "...show that the severity of the <...> increased with the increase in the number..."

Response 8:I have revised the manuscript according to the reviewer's opinion.

Point 9:Please quantify the decrease in the mechanical properties or delete this word.

Response 9:I've removed the the decrease in the mechanical properties from the text.

Point 10:Elastic modulus is not suitable. The material becomes not elastic due to the damage accumulation. A comment (in the text) is necessary.

Response 10:I've commented on the manuscript. In the linear elastic stage, the elastic modulus decreased with the increase in the number of freezing–thawing cycles.

Point 11:Please substantiate the choice of the DEM technique (in the manuscript).

Response 11:I have added two references on discrete element method to my manuscript.

Point 12:The reviewer disagrees to this statement. The contact forces were not measured during the tests. The model can represent tendencies only. How can it help to estimate the damages of the concrete structure? (Please comment that in the text.)

Response 12:By controlling the termination condition of uniaxial compression of concrete numerical model, the contact force between particles at any point on the stress-strain curve is obtained to estimate the damage degree of concrete structure at different points.

Point 13:Too many "parallel bonds" and other repetitions appear. Please improve this paragraph stylistically.

Response 13:I have replaced "parallel bonds" with the abbreviated form Pb.

Point 14:What is the conclusion of Figures 14 and 15? How can the model be used to predict the damage resistance of the concrete?

Response 14:The conclusion of Figure 14 and Figure 15 is that the cracks generated in the uniaxial compression of freezing-thawing concrete mainly occur in the post-peak decline stage. By controlling the termination condition of uniaxial compression of concrete numerical model, the number of cracks at any point on the stress-strain curve is obtained to predict the damage degree of concrete structure at different points.

Point 15:What does it mean? The term "fragment" must be appropriately introduced.

Response 15:The fragment is generated by the penetration of many micro-cracks around the particles. The larger the number of fragments, the more micro cracks around the particles.

Point 16:The numerical model represents the test results adequately. That is an acceptable outcome, but how it can be used in practice? What is the recommendation?

Response 16:The number of concrete fragments and the final failure morphology are used to predict the failure mode and the damage degree of concrete in macroscopic physical test.

Point 17:None of the below statements is a conclusion. They describe the results. What do the authors recommend? What are the mechanisms responsible for the observed outcomes? How can the numerical model used in practice?

Response 17:I have revised the conclusion of the paper.
